# The Complete Mitochondrial Genome of *Stromateus stellatus* (Scombriformes: Stromateidae): Organization, Gene Arrangement, and Phylogenetic Position Within the Suborder Stromateoidei

**DOI:** 10.3390/genes16111256

**Published:** 2025-10-24

**Authors:** Fernanda E. Angulo, Rodrigo Pedrero-Pacheco, José J. Nuñez

**Affiliations:** Instituto de Ciencias Marinas y Limnológicas, Facultad de Ciencias, Universidad Austral de Chile, Valdivia 5090000, Chile; fernanda.angulo@alumnos.uach.cl (F.E.A.); rodrigo.pedrero@alumnos.uach.cl (R.P.-P.)

**Keywords:** mitochondrial genome, molecular characterization, phylogenetic analysis, *Stromateus stellatus*, sustainable management

## Abstract

Background/Objectives: The butterfish *Stromateus stellatus* is undervalued and usually discarded as bycatch, leading to an inefficient and unsustainable use of marine biomass. Overall, although *Stromateus* is the type genus of the family Stromateidae, its species are less studied than more economically important fishes. Methods: In this study, we determined and analyzed the complete mitochondrial genome sequence of *S. stellatus*. Furthermore, we performed maximum likelihood and Bayesian inference analyses to infer the phylogenetic relationships among 21 species of the order Scombriformes. Results: Using next-generation sequencing (NGS) and de novo assembly, a circular mitochondrial genome of 16,509 bp was obtained, exhibiting the typical vertebrate mitochondrial structure comprising 13 protein-coding genes, two ribosomal RNA genes, and 22 transfer RNA genes. Three intergenic regions were identified, including the control region and the origin of light-strand replication, along with several gene overlaps. The heavy strand nucleotide composition was determined to be 28.79% A, 27.84% C, 16.32% G, and 27.05% T, with a GC content of 44.16%. The three *Peprilus* and five *Pampus* species formed a clade together with *S. stellatus,* supported by high bootstrap and posterior probability values, confirming the monophyly of Stromateidae. Conclusions: The gene order and content are consistent with those reported for other Stromateidae species and correspond to the typical arrangement observed in most bony fishes. This mitochondrial genome represents the first one reported for the genus *Stromateus*, providing valuable insights into the genetic makeup of *S. stellatus*, contributing to a better understanding of marine biodiversity. Additionally, these data will support future research on pelagic fish evolution and assist in sustainable fisheries management.

## 1. Introduction

The family Stromateidae Rafinesque 1810 (order Scombriformes), or butterfish, contains 18 fish species divided into three genera: *Pampus*, *Peprilus*, and *Stromateus* [1]. Butterfishes live in coastal waters off North America, South America, West Africa, and the Indo-Pacific region [2], characterized by their small mouth, a forked tail, and a single dorsal fin. Like the related rudderfishes (Centrolophidae) and man-of-war fishes (Nomeidae), stromateids also have peculiar, toothed outpocketings in the esophagus [3].

The genus *Stromateus* comprises three valid species: *S. brasiliensis* (Southwest Atlantic butterfish), *S. fiatola* (blue butterfish), and *S. stellatus* (starry butterfish). Other names historically proposed, such as *S. albus*, *S. simillimus*, *S. capensis*, and *S. microchirus*, are now considered synonyms or have been reassigned to different genera, such as *Pampus*, *Peprilus*, or *Parastromateus* (a Carangiformes genus). Molecular studies identify *Stromateus* as a sister genus to *Pampus*, with both groups likely originating from the ancient Tethys Seaway [4].

*S. stellatus* Cuvier, 1829, is the only stromateid species in the Southeastern Pacific found off the coasts of Perú and Chile [5]. This species forms coastal shoals, often constituting part of the concurrent fauna in the catches of resources such as Peruvian anchoveta (*Engraulis ringens*) and Araucanian herring (*Strangomera bentincki*). Its body is tall, laterally compressed, and grayish, with dark spots above the lateral line, and its head presents a rounded profile. Larval development is well-documented; its size ranges from about 2.7 to 10 mm, and flexion occurs between 5.6 and 7.7 mm, with elongated bodies in early stages. Noteworthy features include three main dorsal dendritic melanophores and specific patterns of ventral and internal pigmentation [6]. In juveniles, pelvic fins are found beneath the thorax, but these fins are absent in adults, which also lack a pelvic spine.

The species is considered inedible due to the presence of diacylglyceryl ether (DAGE) as a major lipid component in its muscle, which can cause intestinal bleeding or diarrhea upon ingestion [7,8]. As a result, *S. stellatus* is undervalued as bycatch and is usually discarded, leading to an inefficient and unsustainable use of marine biomass. Overall, although *Stromateus* is the type genus of the family Stromateidae, its species are less studied than more economically important fishes.

This disparity is primarily due to research and fisheries management efforts prioritizing species that hold high commercial value or are staple food resources in their native regions. For example, *Pampus* species (pomfrets) from the same family are intensely studied and managed due to their high demand in international markets and consistent roles in local fisheries [9,10]. In contrast, *Stromateus* species are less targeted and thus comparatively underrepresented in both the economic and ecological literature. Consequently, the biological, ecological, and taxonomic knowledge of *Stromateus* species lags behind that of its more commercially valuable relatives and other major fishery species, reflecting a global pattern in fisheries research allocation and reporting [11].

The vertebrate mitochondrial genome (mitogenome) is a circular molecule measuring approximately 15–20 kilobases in Animalia [12]. Its predominantly maternal inheritance has a conserved genomic structure, high copy number, lack of introns, and higher mutation rate compared to nuclear DNA [13,14,15]. Moreover, recent advances in DNA sequencing technologies have enabled the rapid and accurate acquisition of complete mitogenome sequences. Consequently, this genetic marker is widely used to resolve taxonomic relationships, reconstruct evolutionary histories, understand genetic diversity and changes in population size dynamics, as well as to identify cryptic species and stock variability [16,17,18]. In marine environments, mitogenomes can provide information on the population structure of commercially valuable species [19], as well as help to develop science-based strategies to manage fisheries more sustainably [20,21].

This study details the genome structure, codon usage, nucleotide composition, and gene order of the mitogenome of *S. stellatus*. To place these findings in context, these features were compared with those of other complete mitogenomes from the family and the suborder Stromateoidei to assess phylogenetic relationships among taxa. This also represents the first mitogenome reported for this genus. As a result, our molecular characterization can serve as a reference for species determination, providing genetic resources for taxonomic, systematic, and genetic research within Stromateidae.

## 2. Materials and Methods

### 2.1. Sample Collection and mtDNA Extraction

An *S. stellatus* specimen was obtained as part of a catch of the artisanal purse-seine fishery of *E. ringens* and *S. bentincki* on 13 May 2024, in the La Barra area, La Araucanía region (39°14′44′′ S; 73°16′53′′ W), Chile. The specimen was lifeless at the time of acquisition, so it was stored directly at 4 °C and subsequently transferred to the facilities of the Austral University of Chile. Liver tissue was homogenized with a Dounce homogenizer under cold conditions using an isotonic Manitol–Sucrose buffer to maintain organelle integrity, as described in Clayton and Shadel’s study [22]. This was followed by low-speed centrifugation to enrich for mitochondrial fractions while eliminating most nuclear DNA. mtDNA was directly isolated from this fraction under standard Phenol/Chloroform DNA extraction protocols and recovered by ethanol precipitation [23]. This enrichment procedure enables cost-effective and in-depth mtDNA sequencing, with sufficient sensitivity to facilitate precise assembly and annotation [24]. Additionally, DNA quality and concentration were determined by 1.5% agarose gel electrophoresis and by fluorometry using a Qubit 4.0 Fluorometer with the Qubit dsDNA HS Assay Kit (Invitrogen, Carlsbad, CA, USA).

### 2.2. Mitochondrial Genome Sequencing, Assembly, and Annotation

The mitogenome sequencing library was prepared following the MGIEasy FS DNA Library Prep Set protocol (MGI Tech, Shenzhen, Guangdong, China) using 200 ng of the isolated DNA. Sequencing of short fragments was performed on the DNBSeq platform (MGI Tech), producing paired-end reads of 150 nucleotides (PE150) on a DNBSeq G400 sequencer.

The quality of the initial sequencing reads was assessed using FastQC v0.12.1 [25]. After quality control, the reads underwent de novo assembly via NOVOPlasty v4.3.5 [26], with assembly parameters set as follows: “Genome Range = 15,000–20,000,” “K-mer = 33,” “Read Length = 151,” and “Insert size = 300.” The *COI* sequence of *S. stellatus* (GenBank accession number AB205450) served as the seed for assembly. To confirm the identity of the assembled sequences as mitochondrial genomes of ray-finned fishes, the resulting FASTA file was searched against the NCBI database using BLASTN+ 2.17.0 [27,28].

The assembled mitogenome sequence was annotated using the MITOS v2.1.7 tool [29] implemented on the Proksee server v1.1.3 (https://proksee.ca/; accessed on 25 March 2025) [30], employing the genetic code “2—Vertebrate Mitochondrial”. A circular mitogenome map was also generated using the Proksee server v1.1.3. The tRNAs’ secondary structures were predicted using the MITOS v2.1.7 tool and further visualized in the Forna web server (force-directed RNA) (http://rna.tbi.univie.ac.at/forna; accessed on 30 March 2025) [31]. The base composition and relative synonymous codon usage (RSCU) of the 13 protein-coding genes (PCGs) were estimated using MEGA 12 [32]. To calculate the nucleotide composition of skew of all 13 PCGs we used AT-skew = (A − T)/(A + T) and GC-skew = (G − C)/(G + C) [33] with the formulae implemented in Excel. The stem-loop secondary structure of the non-coding regions was folded using the UNAfold Server (https://www.unafold.org/mfold/applications/dna-folding-form.php, accessed on 30 March 2025) [34] under the RNA folding option with default parameters.

The *S. stellatus* genome sequence was deposited in the NCBI database under accession number PX223046.

### 2.3. Phylogenetic Analysis

To analyze the phylogenetic position of *S. stellatus*, we downloaded the complete mitogenome sequences of twenty-one scombriform species based on our GenBank BLAST+ 2.17.0 results and the phylogeny proposed by Near and Thacker [35]. The species and their GenBank accession number are shown in Table 1.

For each species, we extracted and concatenated the 13 PCGs into a single matrix. Alignment was performed using MAFFT version 7 [42] with the global pairwise iterative refinement method (G-INS-i). Tree topology was inferred using the maximum likelihood method in IQ-TREE [43,44] and Bayesian inference in MrBayes 3.2.7a [45]. The best-fit substitution models for partitioned data (13 partitions) were selected using ModelFinder 3.0 [46] based on the Bayesian information criterion (BIC). Node support values were assessed by Ultrafast Bootstrap Approximation support (UFBoot%) and posterior probabilities, and the resulting trees were visualized in FigTree v.1.4.5_pre (http://tree.bio.ed.ac.uk/software/figtree/, accessed on 30 March 2025). *B. japonica* (KT908039) and *P. aesticola* (AP012499), both belonging to the family Bramidae, were used as the outgroups. For pairwise genetic distance analyses, we used the same dataset as for the phylogenetic analyses.

## 3. Results

### 3.1. Features and Gene Content of the Mitochondrial Genome of S. stellatus

A total sequencing output of 10,431,628 paired-end raw reads was obtained, of which more than 85% showed, on average, a minimum sequencing quality score of Q30. The mitogenome assembly was built with all reads and had an average coverage depth of 95,781X, which was expected due to the method used, and the high amount of mtDNA obtained.

The complete mitogenome of *S. stellatus* is 16,509 bp in length (Figure 1), comprising the typical structure of 13 protein-coding genes (PCGs), two ribosomal RNA genes, 22 transfer RNA genes, and two non-coding regions (replication origins of the H and L-strands and the control region).

Among the 13 PCGs, only *ND6* is encoded on the L strand, together with eight tRNA genes (*Gln*, *Ala*, *Asn*, *Cys*, *Tyr*, *Ser2*, *Glu*, and *Pro*). The remaining 12 PCGs, 14 tRNA genes, and both ribosomal RNA genes (*12S* and *16S*) are encoded on the H strand (Figure 1; Appendix A). The combined length of all PCGs is 11,431 bp, representing 68.86% of the mitogenome.

The heavy strand nucleotide composition is 28.79% for A, 27.84% for C, 16.32% for G, and 27.05% for T, with a GC content of 44.16% (Appendix A). Within the 13 PCGs, the nucleotide proportions are 26.47% A (3026 bp), 28.45% C (3252 bp), 15.61% G (1784 bp), and 29.47% T (3369 bp). The overall GC content of the PCGs (44.06%) closely approximates that of the mitochondrial genome (44.16%); however, the percentage of thymine (T) is higher in the PCGs (29.47% compared to 27.05%). Among the PCGs, *ND5* is the longest gene (1839 bp), while *ATP8* is the shortest gene (168 bp).

Further analysis revealed that most *S. stellatus* PCGs start with the standard initiation codon ATG, except for the *COI* gene, which begins with GTG. Nine PCGs terminate with the TAA stop codon, while the *COII* and *ND4* genes end with an AAT stop codon and *ND6* and *Cyt b* genes end with a TAG stop codon. Additionally, the nucleotide composition bias predominantly exhibits negative GC and AT skew values (Appendix A).

### 3.2. Codon Usage and Comparative RSCU Among Stromateid Species

The overall codon usage and RSCU for all 13 PCGs in the *S. stellatus* mitogenome, three species of *Peprilus*, and five species of *Pampus* are presented in Figure 2. The total number of codons ranges from 3808 (*P. chinensis*, *P. punctatissimus*) to 3811 (*P. paru*), including the translation termination codons TAA and TAG, and the alternative stop codons AGA and AGG (Appendix A). Comparative RSCU analysis among stromateids indicated that among all 13 PCGs, the codon usage was found to be conserved, with CUU (Leu1), CGA (Arg), and AAA (Lys) representing the most frequently used codons.

### 3.3. rRNA and tRNA Genes of S. stellatus

The mitochondrial genome of *S. stellatus* comprises the small subunit rRNA gene (*12S* rRNA) and the large subunit rRNA gene (*16S* rRNA), which were 950 bp and 1667 bp in size, respectively. These two rRNA genes are separated by *tRNA-Val* (Figure 1). In addition, 22 tRNA genes were detected, interspersed among the rRNA and protein-coding genes, with lengths ranging from 67 to 74 base pairs (bp) (Appendix A). All tRNA genes exhibit the conventional cloverleaf-shaped secondary structure typical of other fish lineages (Appendix A). The most common non-Watson–Crick base pairs found in the tRNA secondary structures are adenine and cytosine (A–C) pairs, which appear in *tRNA-Val*, *tRNA-Trp*, *tRNA*-*Lys*, and twice in *tRNA*-*Ser1*. Thymine and cytosine (T–C) pairs are the next most frequent, found in *tRNA*-*Ile*, *tRNA*-*His*, *tRNA*-*Leu1*, and *tRNA*-*Thr*. Most of these mismatches are in the acceptor and anticodon stems, as shown in Appendix A.

### 3.4. Non-Coding, Overlapping, and Intergenic Regions

The *S. stellatus* mitogenome contains two non-coding regions. The largest non-coding region, the Control Region (CR), is 589 bp long and situated between *tRNA-Pro* and *tRNA-Phe* (positions 15,887 to 16,454; Figure 1; Appendix A). The base composition of this region is 28.69% A, 24.62% C, 16.98% G, and 29.71% T, with a GC content of 41.60%. The CR shows a negative AT-skew of 0.02 and a GC-skew of 0.18 (Appendix A). The other non-coding region includes the origin of L-strand replication (OL) and a second H-strand replication origin (OH1), both located between *tRNA-Asn* and *tRNA-Cys* within the WANCY region (Figure 1). The OL region (32 bp) has a G+C content of 53.13% and forms a stem-loop secondary structure (Figure 3A), while the OH1 region (39 bp) has a G+C content of 61.54% and contains two stem-loop secondary structures (Figure 3B).

Several overlapping and intergenic regions were also observed in the mitogenome. The length of overlapped sequences ranges from 1 bp to 25 bp (Appendix A), with the largest overlapping region was located between the O_L_ and O_H1_ regions. A 10 bp overlap containing the motif “ATGACACTAA” is found between *ATP8* and *ATP6*. In addition, a 7 bp overlap with the motif “ATGCTAA” is detected between *NAD4L* and *NAD4*. The largest intergenic region (205 bp) lies between *tRNA-Pro* and the Control Region (positions 15,660 to 15,866; Appendix A). This is followed by a 55 bp intergenic region between the Control Region (position 16,454) and *tRNA-Phe* (position 1), and a third notable intergenic region of 27 bp is located between positions 1090 (*tRNA-Val*) and 1117 (*16S* rRNA).

### 3.5. Mitochondrial Gene Rearrangement in Stromateidae

The comparative mitochondrial gene arrangement of *S. stellatus* relative to other stromateids is presented in Figure 4. The positions and orientations of protein-coding genes, tRNA genes, and rRNA genes, as well as intergenic and overlapping regions, are highly conserved across the family Stromateidae. The only notable exception is the size of the Control Region and the origin of L-strand replication, which, in *Pampus cinereus,* is situated on the L-strand [47].

### 3.6. Phylogenetic Relationships

Phylogenetic analyses were conducted on a concatenated dataset of 13 PCG sites. This dataset contained 11,547 nucleotide-aligned sites, of which 5198 were parsimony-informative, 1016 were singletons, and 5333 constant sites. Both analyses, maximum likelihood (Figure 5) and Bayesian Inference (Appendix A), strongly support *S. stellatus* as the sister taxon to species of the genus *Peprilus*. Along with *Pampus*, these taxa form a well-supported clade, confirming the monophyly of the family Stromateidae. The families Ariommatidae and Nomeidae are identified as close relatives within the Stromateidae clade. Additionally, the families Amarsipidae, Tetragonuridae, Ariommatidae, Nomeidae, and Stromateidae together form a relatively well-supported clade, endorsing the monophyly of the suborder Stromateoidei. Notably, the family Scombrolabracidae, currently classified within suborder Scombroidei, was placed with relatively good support in a clade alongside Centrolophidae within suborder Stromateoidei.

## 4. Discussion

In the present study, the main features of the complete mitochondrial genome of the butterfish *S. stellatus* are described and compared with previously published butterfish mitochondrial genomes. This is also the first full and well-characterized mitogenome presented for the genus *Stromateus*, which helps clarify the systematics of Stromateidae members and facilitates molecular identification.

The *S. stellatus* mitogenome consists of 37 genes, including 13 protein-coding genes, 22 tRNA genes, two rRNA genes, and the Control Region. The gene structure and layout in this mitogenome, particularly the organization of protein-coding, tRNA, and rRNA gene blocks, are identical to those in other stromateids, such as *Peprilus* and *Pampus* [10,37,39,40], and stromateoids, including *Ariomma*, *Cubiceps*, and *Tetragonurus* [37,38]. In all cases, the genes are arranged in the same order and orientation, with no signs of gene rearrangements (Figure 4). Consistent with the typical tRNA set found in actinopterygian mitogenomes, 22 tRNA genes were predicted, including two types each of *tRNA-Leu* and *tRNA-Ser* (Appendix A). All predicted tRNAs have the classic clover-leaf secondary structure, including *tRNA-Ser1* (AGY), which, among fish mitogenomes, often fails to form a stable structure due to the absence of the DHU arm [48,49,50]. In *S. stellatus,* such as in nearly all vertebrate mitogenomes, the origin of light-strand replication (O_L_) is located within the “WANCY” region, which contains five tRNA genes: *tRNA-Trp*, *tRNA-Ala*, *tRNA-Asn*, O_L_, *tRNA-Cys*, and *tRNA-Tyr* (WAN-O_L_-CY) [51,52].

The *S. stellatus* mitogenome also contains three extensive intergenic regions and several long gene overlaps. Specifically, the largest intergenic region measures 205 base pairs, followed by regions of 55 and 27 base pairs, the latter located between the *tRNA-Val* and *16S* rRNA. Similar intergenic regions are also present in the mitogenomes of *Pampus* and *Peprilus* species, suggesting a conserved pattern across related taxa. These sequence motifs are also conserved among vertebrates [53,54]. The findings indicate that the mtDNA of *S. stellatus,* as well as other members within Stromateidae and the order Scombriformes, exhibits a higher degree of gene structure conservation; consequently, they are of great significance for further exploration of species boundaries and divergence times.

The comparison of the RSCUs of the *S. stellatus* mitogenome with other stromateid mitogenomes is shown in Figure 2. The Leu1, Val, Ser2, Pro, Thr, Ala, Arg, and Gly amino acids utilize four different codons in all species, while two codons encode all other amino acids, consistent with other fish lineages [55]. An RSCU value below 1.0 indicates a negative codon bias, meaning the codon is used less frequently than expected. On the contrary, an RSCU value greater than 1 indicates that the codon is used more frequently than expected, suggesting a positive codon usage bias toward that codon. Although the mitochondrial genetic code in animals is conserved, significant variations in synonymous codon usage have been observed among and within species [56]. Codon usage bias can influence gene expression, protein folding, tRNA abundance, and genome evolution. For example, genes showing strong codon bias often have characteristic RSCU profiles with codons favored for translational efficiency [57]. Furthermore, the structure, function, and expression of proteins may be impacted by the choice of codons used [58]. This choice is crucial for translational efficiency, accuracy, and protein synthesis, as different organisms opt for codons that can be quickly processed to minimize the duration and energy required for translation.

In all PCGs (except *NAD6*), the leading coding strand displays negative GC-skew, meaning it is richer in cytosine than guanine. Negative GC-skew in a mitochondrial genome implies that in a specific strand of the mitochondrial DNA, the amount of cytosine (C) nucleotides is higher than the amount of guanine (G) nucleotides. This asymmetry arises due to differences in DNA replication and mutational or selective pressures on the leading and lagging strands. In other words, it is related to the replication mechanism, where one strand is synthesized continuously (the leading strand) and the other discontinuously (the lagging strand), resulting in compositional biases between the strands.

The phylogenetic reconstruction in this study was based on the sequences of 13 protein-coding genes from 22 mitochondrial genomes, including nine from Stromateidae. The topology of the resulting phylogenetic trees was supported by high bootstrap and posterior probability values (Figure 5), with the three *Peprilus* and five *Pampus* species forming a monophyletic group together with *S. stellatus*. Although [59] regarded *Stromateus* as sister species of *Pampus*, the phylogenetic inference presented here indicates that *Peprilus* is more closely related to *Stromateus* than to *Pampus*. Similar results were obtained by Doiuchi et al. [59] and Pastana et al. [3] based on morphological characters. However, due to the limited number of species included in these analyses, systematic conclusions within the suborder Stromateoidei should be interpreted with caution.

The increasing availability of published mitochondrial genomes, the conserved structure of these genomes in vertebrates, and advances in assembly methods have enhanced the value of mitochondrial genomes in taxonomic and evolutionary research. In fact, many marine organisms, such as cnidarians, crustaceans, and fish, are considered sentinels of environmental health and ecosystem stability. Thus, in the case of fish, establishing reference databases of mitochondrial genomes can significantly enhance monitoring through environmental DNA (eDNA). Furthermore, as point out in previous studies [56,60], this trend aids in the development of evidence-based conservation strategies, especially for taxa previously neglected due to their low market value.

*S. stellatus* is the sole stromateid in the coastal waters of Perú and Chile, yet its economic undervaluation due to its inedibility has led to it being classified as a bycatch species, resulting in considerable waste and unsustainable marine biomass use. Although due to sample availability, we did not perform population analyses to determine the genetic diversity of this species, our contribution of the first *Stromateus* mitogenome supplies a valuable genetic resource for further taxonomy, phylogeny, and conservation research within Stromateidae. Moreover, though population structure assessment awaits broader sampling, the recognition of research bias toward commercial species reiterates the need for more inclusive scientific approaches and management policies, promoting the sustainable use of marine resources by integrating neglected species into research and monitoring.

## 5. Conclusions

In this work, the complete sequence of the mitochondrial genome of the butterfish *S. stellatus* is presented for the first time. Its basic characteristics show a typical genome organization and gene order found in other Scombriformes and Actinopterigii mitochondrial genomes. Phylogenetic analysis, based on the 13 PCGs, supports the monophyly of Stromateidae and indicates a close phylogenetic relationship between *Stromateus* and *Peprilus*. Overall, the mitochondrial genome of *S. stellatus* constitutes a data resource that advances the systematics of Stromateidae, establishing a basis for future molecular and ecological studies.

## Figures and Tables

**Figure 1 genes-16-01256-f001:**
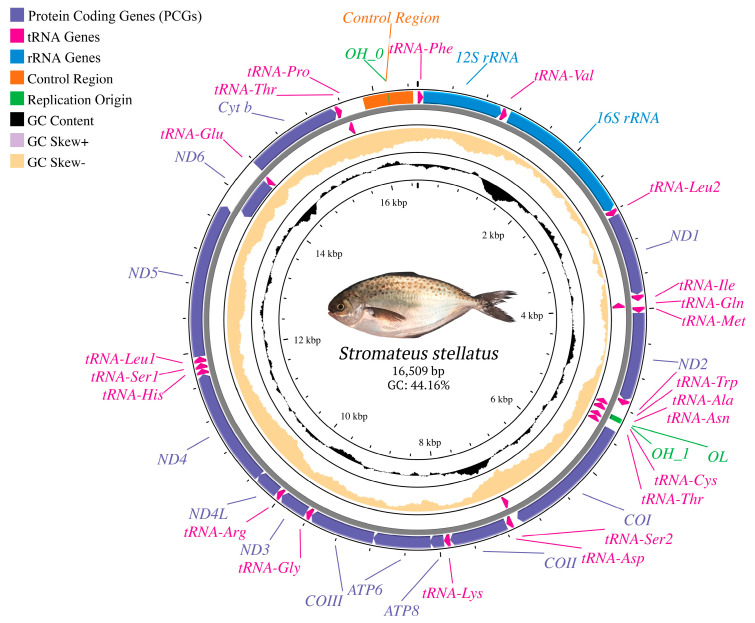
The mitogenome map of *S. stellatus*. The arrows indicate the gene transcription orientation. The tRNA genes are labeled according to their corresponding amino acid codes. The color codes for different gene types are shown in the legend on the map.

**Figure 2 genes-16-01256-f002:**
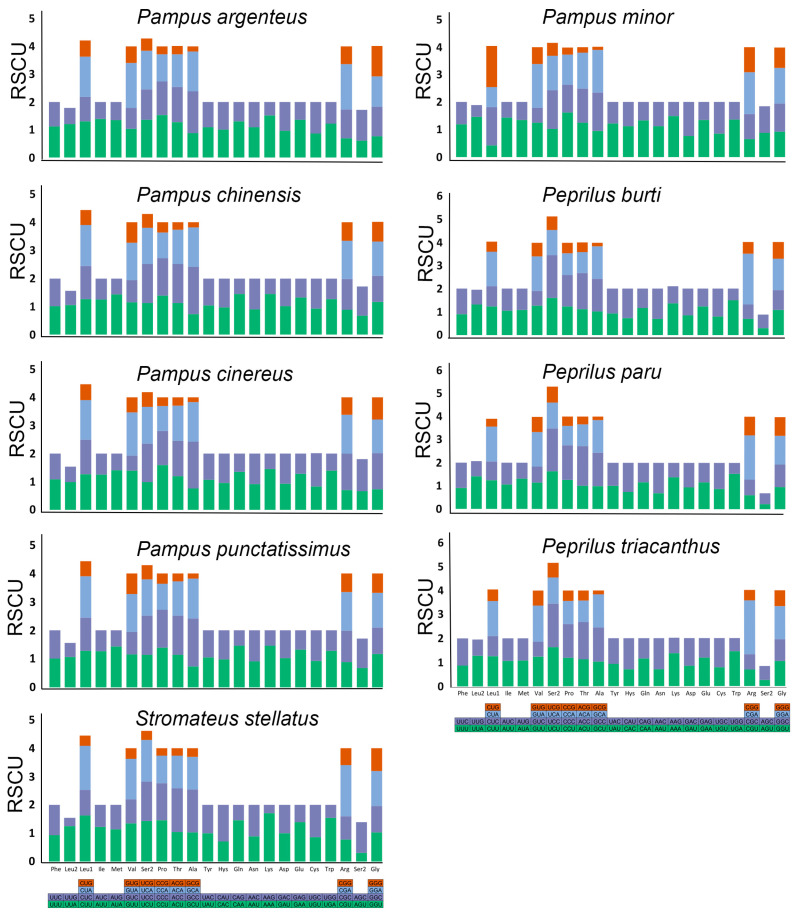
Relative synonymous codon usage (RSCU) in the 13 PGSs of the mitochondrial genome of *S. stellatus* and eight mitogenomes from the family Stromatidae: *P. argenteus* (KF373560), *P. minor* (MH037007), *P. cinereus* (OR538383), *P. chinensis* (KJ418377), *P. punctatissimus* (OR538387), *P. paru* (OP056882), *P. burti* (AP012947), *P. triacanthus* (AP012518).

**Figure 3 genes-16-01256-f003:**
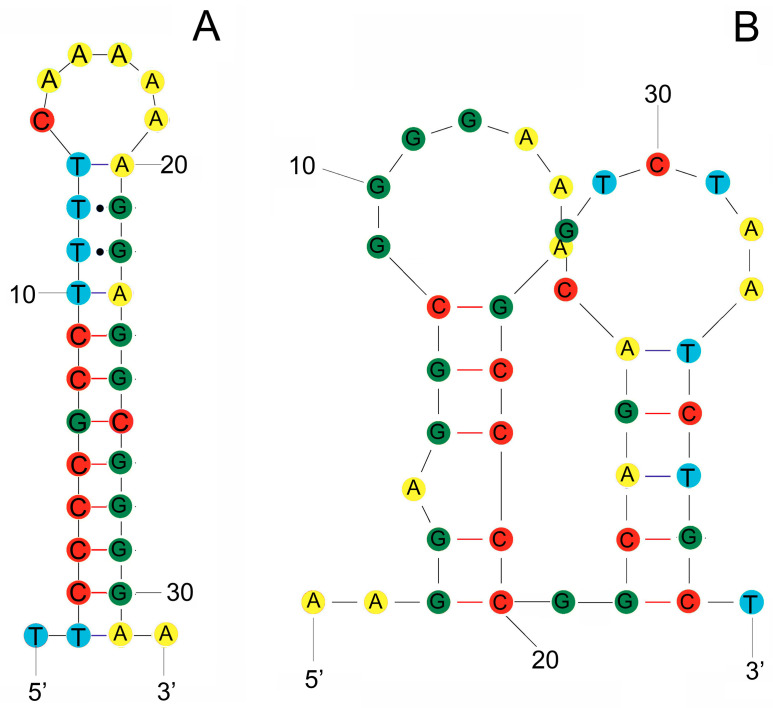
Stem-loop secondary structures of two non-coding regions in the *S. stellatus* mitogenome. (**A**) Stem–loop secondary structure of the O_L_ region (32 bp); (**B**) Stem–loop secondary structure of the O_H1_ region (39 bp). Red and black bonds indicated GC and TA pairs, respectively. The dots indicate mismatches between bases.

**Figure 4 genes-16-01256-f004:**
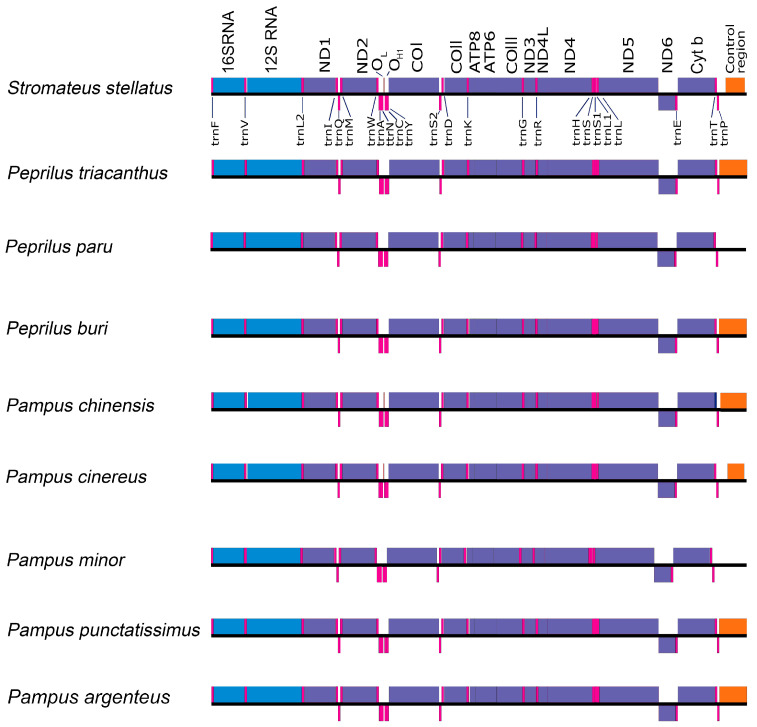
Comparative mitogenome organization of *S. stellatus* and eight Stromateidae species. The colors show what is indicated in Figure 1.

**Figure 5 genes-16-01256-f005:**
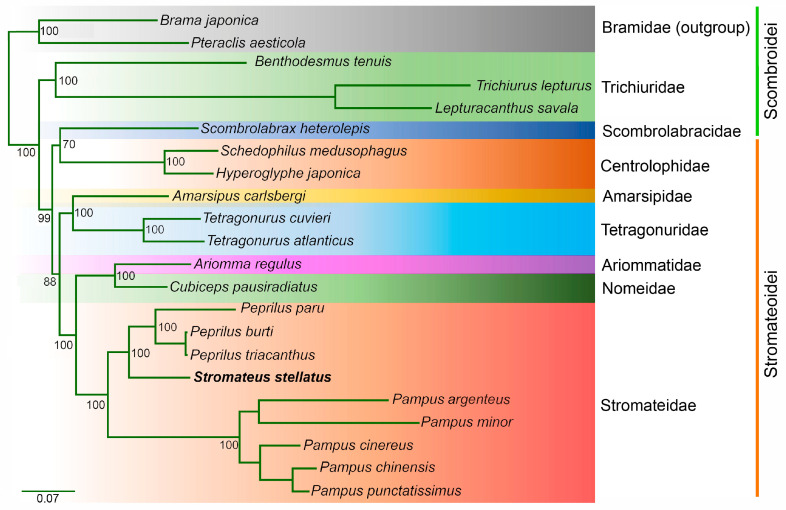
The maximum-likelihood (ML) phylogenetic tree of *S. stellatus* and 21 Scombriformes species. Phylogenetic reconstruction was performed from a concatenated matrix of 13 protein-coding mitochondrial genes and 11,547 aligned sites. The numbers at the nodes indicate UFBoot support values.

**Table 1 genes-16-01256-t001:** List of mitogenomic data of Scombriformes species from GenBank used for the maximum likelihood and Bayesian phylogenetic analyses.

Family	Species Name	GenBank Accession Number	References
Amarsipidae	*Amarsipus carlsbergi*	NC_037474	[36]
Ariommatidae	*Ariomma regulus*	PV357913	Unpublished
Bramidae	*Brama japonica*	KT908039	Unpublished
	*Pteraclis aesticola*	AP012499	[37]
Centrolophidae	*Schedophilus medusophagus*	MT410878	Unpublished
	*Hyperoglyphe japonica*	AP006037	[38]
Nomeidae	*Cubiceps pausiradiatus*	AP006038	[38]
Tetragonuridae	*Tetragonurus cuvieri*	AP012514	[37]
	*Tetragonurus atlanticus*	AP012515	[37]
Trichiuridae	*Benthodesmus tenuis*	AP012522	[37]
	*Lepturacanthus savala*	OP724236	Unpublished
	*Trichiurus lepturus*	NC_018791	Unpublished
Scombrolabracidae	*Scombrolabrax heterolepis*	OP035059	Unpublished
Stromateidae	*Pampus argenteus*	KF373560	[39]
	*Pampus minor*	MH037007	[40]
	*Pampus cinereus*	OR538383	[10]
	*Pampus chinensis*	KJ418377	[41]
	*Pampus punctatissimus*	OR538387	[10]
	*Peprilus paru*	OP056882	Unpublished
	*Peprilus burti*	AP012947	[37]
	*Peprilus triacanthus*	AP012518	[37]

## Data Availability

The mitochondrial genome sequence data are openly available from the NCBI GenBank database under accession number PX223046.

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
