# Peer review of "The Complete Mitochondrial Genome of Stromateus stellatus (Scombriformes: Stromateidae): Organization, Gene Arrangement, and Phylogenetic Position Within the Suborder Stromateoidei"

_genes, 2025, doi:10.3390/genes16111256_

Round 1

Reviewer 1 Report

Comments and Suggestions for Authors

For the author comments, please see the attached PDF file. Thank you. 

Author Response

Dear reviewer,

Please find attached the responses to your valuable and constructive comments. The points and comments you considered, as well as those of the other reviewers, significantly improved our work.

Reviewer 2 Report

Comments and Suggestions for Authors

     The mitogenome of butterfish Stromateus stellatus is unknown. Angulo et al. sequenced, assembled, and characterized its mitogenome. The methods are established, were correctly applied, and the results reported appropriately. My comments are minor and aimed at polishing the manuscript. I offer some comments here and also provide a marked manuscript to guide revision.      

     Introduction. – At line 49, the correct common name of Engraulis ringens is Peruvian anchoveta. At line 50, the correct common name of Strangomera bentincki is Araucanian herring.

     Methods. – At line 88, was the Qubit fluorometer produced by Invitrogen?

     At line 126, the algorithm referred to is Maximum Likelihood.

     At line 128, the acronym PGCs should be defined for the reader.

     Results. – Figures 3 and 5 can be enlarged to the full width of the page to improve readability.

     Discussion. – I have marked a passage at lines 290-292 that can and should be removed. We will not value a bycatch species that is inedible, and the passage is not really needed.

     At line 344, I suggest addition of “are inherited as a unit and thus comprise” as marked on the manuscript.

     “Significant” should be stricken at line 364. That judgement should be left to the reader.

     The declarations passages fail to mention a scientific collection permit or an animal care and use permit. If these were not needed, the authors should state that.

     References. – I’ve marked stylistic issues in the literature citations, mostly pertaining to inconsistent use of capitalization for article titles.

Comments on the Quality of English Language

The English prose can be polished with attention to the comment written directly upon the manuscript. 

Author Response

(The authors gave the same response as above.)

Reviewer 3 Report

Comments and Suggestions for Authors

The manuscript presents the complete characterization of the mitochondrial genome of Stromateus stellatus, providing valuable insights into the systematics of Stromateidae and contributing data that can support sustainable fisheries management. I find the study relevant and of interest to the scientific community, particularly given the limited genomic resources available for this family.

However, several issues should be addressed before the manuscript must be corrected:

As major red flags:

The description of stop codons has to be incorrect, since AAT and CTT are not valid termination codons in the vertebrate mitochondrial code: AAT encodes asparagine and CTT encodes leucine. Sometimes incomplete codons T- or TA- can occour in mitocondrial DNAs and that might be the case here. If that were to be the case, the genbank annotation must be corrected. PLease, review this.

In addition, and probable partly related, the gene lengths reported in Table 3 are necessarily incorrect. For example, the lengths given for COII (691 bp), ND3 (349 bp), ND4 (1381 bp), and Cyt b (1141 bp) are not divisible by three, which is inconsistent with protein-coding sequences. This indicates either a wrongly performed or interpreted annotation for the mitocondrial genoma of this species. Please, take more care.

Based on the figures presented, I noticed that some information is missing from the footnotes and legends. For example, in Figure 4a, what does the white “C” represent? Why are some nucleotide pair bonds shown in red while others are black? In Figure 1, why is the control region depicted in green when the legend indicates it should be orange? Additionally, what do the colors in Figure 5 signify, and why do some of these not correspond to the color scheme used in Figure 1? Some of these questions may be obvious, but they reflect dubious/inacurate representations.

Moreover, the authors state that they conducted a Bayesian inference in MrBayes on a concatenated alignment including all PCGs. NO problems here. However, they also indicate that a single substitution model (TIM2+F+I+G4) was used. It is important to clarify whether the Bayesian analysis was performed using partitions for the individual PCGs, which is generally the recommended way to go, or if the analysis was conducted under a single model without partitioning, as this can affect the accuracy of the phylogeny.

In conclusion, while the study addresses a note on the mitochondrial genome of Stromateus stellatus, several substantial issues need to be carefully addressed. I recommend that the authors revise these points thoroughly to ensure the accuracy, clarity and reliability of their manuscript.

Comments on the Quality of English Language

As minor comments, I happenend to find some typos and style issues, including (but not limited to):

Line 15: “we perform a maximum likelihood” → should read “we performed” (verb tense).

Line 23: “formed a clade together S. stellatus” → should read “formed a clade together with S. stellatus”.

Line 60: “Its predominantly maternal inheritance, has…” → remove the comma after inheritance.

Line 229: “GC-skew 0.18” → should be “GC-skew = 0.18” for consistency.

Line 381: “We want to thanks to LM GOJUMAR I ship” → should read “We want to thank the LM GOJUMAR I ship”.

Author Response

(The authors gave the same response as above.)

Round 2

Reviewer 1 Report

Comments and Suggestions for Authors

Dear authors, 

please find attached the minor comments on your revised manuscript. These comments mainly refer to writing and phrasing the added text sections. 

Best regards

Author Response

Dear reviewer, I am attaching our responses to your comments. 
